# Role of Viral Ribonucleoproteins in Human Papillomavirus Type 16 Gene Expression

**DOI:** 10.3390/v12101110

**Published:** 2020-09-30

**Authors:** Naoko Kajitani, Stefan Schwartz

**Affiliations:** Department of Laboratory Medicine, Lund University, 22184 Lund, Sweden; stefan.schwartz@med.lu.se

**Keywords:** human papillomavirus (HPV), papillomavirus, SR proteins, hnRNP, splicing, polyadenylation

## Abstract

Human papillomaviruses (HPVs) depend on the cellular RNA-processing machineries including alternative RNA splicing and polyadenylation to coordinate HPV gene expression. HPV RNA processing is controlled by cis-regulatory RNA elements and trans-regulatory factors since the HPV splice sites are suboptimal. The definition of HPV exons and introns may differ between individual HPV mRNA species and is complicated by the fact that many HPV protein-coding sequences overlap. The formation of HPV ribonucleoproteins consisting of HPV pre-mRNAs and multiple cellular RNA-binding proteins may result in the different outcomes of HPV gene expression, which contributes to the HPV life cycle progression and HPV-associated cancer development. In this review, we summarize the regulation of HPV16 gene expression at the level of RNA processing with focus on the interactions between HPV16 pre-mRNAs and cellular RNA-binding factors.

## 1. Introduction

Human Paillomaviruses (HPVs) are small DNA viruses that infect cutaneous or mucosal epithelial keratinocytes [1,2]. HPVs are transmitted by contact and/or sex, but the majority of all HPV infections are asymptomatic and are cleared spontaneously [3,4]. In rare cases, infections with high-risk HPVs, e.g., HPV16 and HPV18, persist and may cause anogenital cancer, primarily cervical cancer and anal cancer, or head and neck cancer, primarily tonsillar cancer [3]. HPV is strictly epitheliotropic, and the HPV life cycle and gene expression program are intimately linked to the differentiation state of the infected keratinocyte. Replication of the HPV genome requires cell mitosis, whereas production of progeny virus is dependent on terminal cell differentiation [5]. In rare cases, HPV-infected cells fail to differentiate terminally and remain under control of the HPV E6 and E7 oncogenes that promote mitosis and inhibit apoptosis [6]. These HPV-infected cells do not progress to the late stage of the HPV life cycle and represent a state of nonproductive persistence that increases the risk for cancer development [7,8].

Control of HPV gene expression is of paramount importance in the viral life cycle as well as in the rare event of carcinogenesis. Although transcriptional regulation of HPV gene expression plays a major role during the life cycle, HPV has relatively few promoters [9,10], which limits the ability to fine-tune control of HPV gene expression at the transcriptional level. However, the pre-mRNAs produced by HPV contain a number of splice sites and two polyadenylation signals that are used to process HPV mRNAs into a myriad of alternatively spliced mRNAs with distinct coding potential [11,12,13,14]. Thus, control of splicing and polyadenylation is particularly important in the HPV life cycle and pathogenesis since it allows for temporal and spatial regulation of HPV gene expression in a cell differentiation-dependent manner. Formation of ribonucleoprotein complexes that consist of HPV mRNAs and viral and/or cellular proteins plays an essential role in the HPV gene expression program that is constantly responding to the changing intracellular environment in the HPV-infected keratinocyte.

Here, we review the regulation of HPV mRNA splicing and polyadenylation with focus on individual HPV splice sites and polyadenylation signals and the cellular and viral factors that control these sites. In addition, we discuss the connection of these HPV RNA-processing events with transcription, RNA modifications, epigenetics, and the DNA damage machinery. Multiple previous reviews have covered RNA processing in human as well as bovine papillomavirus [11,13,15,16,17]. In this current review we will focus primarily on human papillomavirus type 16 (HPV16), but other papillomaviruses are discussed when appropriate. Finally, we identify areas of research that would be of interest to explore further.

## 2. HPV Life Cycle

HPV is a small DNA virus with an approximately 8 kb circular, double-stranded DNA molecule as a genome [18,19] (Figure 1). The genomic structure is divided into three regions: Long control region (LCR), early gene region (E6, E7, E1, E2, E4, and E5), and late gene region (L2 and L1) (Figure 1 and Figure 2). The HPV LCR contains a replication origin (Ori) on which HPV replication factors E1 and E2 bind and initiate replication of the HPV DNA [7]. The HPV LCR also harbors the HPV early promoter, whereas the more elusive late promoter is located in protein-coding regions further down on the HPV genome. In addition to these two promoters, HPVs have two polyadenylation signals, named early (pAE) and late (pAL) polyadenylation signal (Figure 1). The early promoter and pAE are active during the early stage of the viral life cycle, whereas the late promoter and pAL are active at the late stage. Transcription from the early promoter normally terminates at the pAE, and transcription from the late promoter at the pAL, suggesting that there is one pre-mRNA species for early and one for late [13] (Figure 2, “early pre-mRNA” and “late pre-mRNA”). However, it should be noted that many mRNAs initiated at the late promoter are polyadenylated at pAE (transcripts J and K) (Figure 2), and some mRNAs initiated at the early promoter are polyadenylated at pAL (transcripts G, H, and I) (Figure 2).

HPV pre-mRNAs are synthesized by cellular RNA polymerase II (Pol II). Transcription initiation is followed by RNA processing that includes 5′-cap attachment, alternative RNA splicing, and polyadenylation. Translation of HPV mRNAs is conducted by the cellular translation machinery in a cap-dependent manner. Since the majority of the HPV pre-mRNAs/mRNAs are structurally polycistronic [20,21] (Figure 2), it is important that a variety of alternatively spliced mRNAs are generated to ensure that each HPV ORF is in a position that is favorable for translation on at least one mRNA species. Thus, alternative splicing of HPV mRNAs does not only offer opportunities for regulation of HPV gene expression, it is also necessary to generate mRNAs that can be efficiently translated into all HPV proteins [13].

HPV life cycle progression is coupled to host epithelial cell differentiation [22]. Since HPV lacks a DNA polymerase, it is completely dependent on the DNA replication machinery of the host cell [23]. Initially, HPVs infect only basal cells on the basement membrane of the stratified epithelium since these cells express the HPV receptors [24]. In addition, mitotic cells on the epithelial basement membrane also express cellular DNA polymerase that is required for replication of the HPV DNA genome. Following HPV entry into the basal keratinocyte, the HPV DNA genome is transported into the nucleus where it is amplified to 50–100 copies per cell [25]. These viral genomes are segregated into mother and daughter cells during cell division. Once the daughter cells detach from the epithelial basement membrane due to vertical apical cell division, these cells start to differentiate and shut down the cellular DNA replication machinery. However, the DNA replication machinery of the cell is responsible for HPV DNA replication. In response to these events, HPV proteins E6 and E7 target cellular antitumorigenic proteins p53 and pRb, respectively, to reactivate the cellular DNA replication machinery and to avoid apoptosis induction [7,26,27]. HPV early proteins E1 and E7 also activate the DNA damage response (DDR) that contributes to HPV DNA synthesis in differentiated cells [28]. The HPV proteins E1 and E2 are key players in HPV DNA replication as E1 functions as DNA helicase and E2 is an HPV DNA-binding protein that recruits HPV E1 and the cellular DNA polymerase to the origin of replication (ORI) in the LCR of the HPV genome [29,30].

In transition to the late stage of the HPV life cycle, the expression levels of E1 and E2 increase, enabling vegetative amplification of the HPV DNA in the differentiated epithelial cell layers [18]. The vegetative HPV DNA replication is immediately followed by the expression of viral late capsid proteins, L1 and L2 [5], and the assembly of viral particles that consist of HPV DNA and the HPV L1- and L2-casid proteins. The virions are released through spontaneous epithelial cell shedding at the uppermost epithelial cell layers. Although the function of the HPV proteins E1^E4 and E5 remains enigmatic, E1^E4 is believed to play a role in the HPV late infection phase as it rearranges cellular cytoskeleton to enhance virion release [31], whereas E5 appears to aid in the evasion of the immune system via affecting the endocytic trafficking as well as to increase the cellular proliferation capacity [32,33].

## 3. HPV16 Gene Regulation

### 3.1. The Switch from HPV16 Early to Late Gene Expression

The HPV16 promoter-switch is the first step in the activation of HPV16 late gene expression, but the promoter-switch is not sufficient for spatiotemporal production of all the HPV16 mRNAs that are needed at this stage in the HPV life cycle. Regulated changes in HPV16 mRNA alternative splicing and polyadenylation are equally important [13]. An HPV16 transcript map has been established based on mRNAs detected in the W12 cell line which is derived from a low-grade lesion and contains nonintegrated episomal HPV16 DNA [20]. A number of HPV16 transcripts encoding E6, E6*I, E6*II, E6^E7, E7, E1, E1C, E2, E1^E4, E5, L2, and L1 were identified, and 5′-splice sites and 3′-splice sites were identified by sequencing of cDNAs representing alternatively spliced HPV16 mRNAs. A complete map of the HPV16 mRNAs and splice sites can be found at the Papillomavirus Episteme website (PaVE) (https://pave.niaid.nih.gov/#home). We have listed some of the alternatively spliced mRNAs that are key to the discussion in this review (Figure 2).

What is the purpose of the switch from the HPV16 early to late promoter? Looking at the HPV16 promoter locations (Figure 2), the most obvious reason for the switch from the HPV16 early promoter (P97) to the HPV16 late promoter (P670) is to bypass transcription of the E6 and E7 ORFs. As the HPV16 E7 mitotic driver fades away, cell differentiation is restored, which in turn activates the HPV16 late promoter. Despite the fact that multiple transcription factors that bind the HPV16 early promoter and enhancer have been identified, little is known about the more elusive late promoter [10,34]. It remains to be determined if there is a connection between the HPV promoters and their transcription factors with factors of the splicing machinery.

A change in HPV16 promoter usage does not entirely explain how expression of the HPV16 late L1 and L2 genes is activated as both early and late promoters have the potential to express L1 and L2 mRNAs (Figure 1 and Figure 2). The HPV promoter-switch is induced by reduced levels of cellular transcription factors which activate the early promoter and high levels of E2 that inhibit the early promoter as cells differentiate, which in turn activates the HPV late promoter [35,36]. To express late L1 and L2 mRNAs, this switch must be accompanied by a reduction of polyadenylation efficiency at pAE, or all mRNAs would continue to be cleaved and polyadenylated at pAE, thereby effectively preventing L1 and L2 expression [37]. Thus, the major role for pAE is to prevent premature L1 and L2 expression at the early stage of the HPV life cycle [38,39]. Activation of the late promoter and inhibition of the pAE pave the way for production of L1 and L2 mRNAs that are polyadenylated at the late polyA signal, pAL. However, one may envisage a transition period in which both promoters and/or both polyA signals are active, resulting in mRNAs initiated at the early promoter and polyadenylated at pAL, and mRNAs initiated at the late promoter and polyadenylated at pAE. As a matter of fact, the identification of such mRNAs bears witness of an early-to-late transition period in the life cycle of HPV [21,40] (for example, transcripts G, H, I, J, and K in Figure 2), but the significance of these mRNAs and the transition stage is unknown.

### 3.2. Exons and Introns on HPV16 mRNAs

Exons and introns are defined primarily by studies of cellular mRNAs [41,42]. Splicing regulatory RNA elements are recognized by splicing factors such as serine/arginine (SR)-rich protein family or heterogeneous nuclear ribonucleoprotein (hnRNPs), modulating the recruitment of spliceosomal components to the pre-mRNA to enhance or interrupt exon/intron definition. Although other RBPs have been implicated in influencing splicing, SR proteins and hnRNPs are thought to be the major RBPs regulating splicing [43]. An exon is a part of a gene that is present on the mature mRNA and that often codes for an amino acid sequence. The vast majority of all cellular genes are broken up by DNA sequences called introns, leaving the shorter exons dispersed between longer introns on a gene. To synthesize a functional mRNA that can be translated into a functional protein, the dispersed exons need to be joined and introns excised in an RNA splicing process. The genomic organization of the cellular genome differs from the organization of the HPV genome in that exons in the cellular genome are rare and often short (average 160 bp [44]). As a consequence, the cellular pre-mRNAs are totally dominated by intronic sequences (average 6938 bp [44]), rendering the identification of the short exons embedded among the long introns particularly challenging during mRNA maturation. In contrast, the HPV16 genome is almost entirely covered by ORFs or protein-coding sequences (CDSs), many of which are overlapping (Figure 2). As a matter of fact, 92% of the HPV16 genome is protein coding, and the average exon size is 1004 bp, and the average intron size is 2285 bp. Although the HPV16 mRNA splicing process has to be as accurate as the splicing of cellular mRNA, the definition of exons and introns in HPV16 may differ to some extent since HPV exons are significantly bigger and more densely packed than cellular exons (Figure 1 and Figure 2). As a consequence, many of the HPV coding sequences serve as both exons and introns, e.g., the E6, E1, and L2 ORFs are good examples (Figure 2). Thus, the E6, E1, and L2 ORFs are defined as exonic regions that must be included in the mature mRNA in order to produce fully functional E6-, E1- or L2-producing mRNAs (for example, mRNAs A, C, G, L, and M in Figure 2), while the same ORFs must be defined as introns in order to generate the spliced E7-, E2- or E1^E4-, and L1-producing mRNAs (for example, mRNAs B, D, E, H, I, N, and O in Figure 2). Therefore, processing of HPV16 pre-mRNAs is a challenge that differs somewhat from the processing of cellular mRNAs. Regulation of HPV RNA processing executed by RNA-RNA-binding proteins (RBP) interactions plays a key role in the HPV gene expression program and requires exceptional precision to properly select splice sites and polyA signals following transcription. Cellular RBPs may contribute to control of RNA processing at multiple levels (Figure 3).

Processing of HPV pre-mRNAs includes and relies on alternative splicing and polyadenylation and is dependent on the cellular splicing and polyadenylation machineries. Outcome of alternative splicing or polyadenylation is influenced by several factors: (1) the strength of splice sites or polyadenylation (polyA) signals; (2) the presence of cis-acting regulatory RNA sequences on pre-mRNAs; and (3) the expression levels of various trans-acting factors, e.g., RBPs [45,46,47]. Strong splice sites possess consensus sequences that are efficiently recognized by the splicing machinery, thus leading to constitutive splicing. Weak splice sites differ from the consensus sequence and their recognition by the splicing machinery is highly dependent on the presence of cis-acting regulatory sequences located adjacent to splice sites. The activity of the regulatory RNA elements depends on the intracellular millieu defined by levels and activity of trans-acting factors such as RBPs. All HPV16 splice sites deviate from consensus sequences and are weak by definition [42] (Table 1). Thus, HPV16 has evolved to contain weak splice sites that can be regulated. Consequently, there are no constitutively active splice sites on HPV16 mRNAs.

## 4. Regulation of HPV16 Gene Expression Mediated by Interactions between HPV16 RNA and Cellular Proteins

### 4.1. Alternative Splicing and Polyadenylation

#### 4.1.1. HPV16 Splice Sites

The HPV16 genome encodes at least four splice donors (SD or 5′-splice sites) (SD226, SD880, SD1302, and SD3632) and seven splice acceptors (SA or 3′-splice sites) (SA409, SA526, SA742, SA2582, SA2709, SA3358, and SA5639) (Figure 2). The consensus sequence of a splice donor is (C/A)AG_**GU**(A/G)AGU, whereas the splice acceptor is (C/U)nX(C/U)**AG**_(A/G) (the most conserved nucleotides are in bold face) [42] (Table 1). It appears that all HPV16 splice donors are suboptimal and deviate from the consensus, although the sequences immediately flanking the cleavage position (AG_**GU**A) are highly consistent with consensus sequence (Table 1).

Among HPV16 splice donors, SD3632 appears to conform better to the consensus splice donor than the remaining HPV16 splice sites. However, HPV16 splice donor SD3632 is strongly suppressed and exclusively used for HPV16 late mRNA splicing, suggesting that this splice site is silenced by cis-acting regulatory RNA elements and/or trans-acting factors. Regarding HPV16 splice acceptors, the sequences flanking the cleavage position of the splice sites are diverse and contribute to the strength of each splice site (Table 1). The consensus 3′-splice site ends introns with an almost invariable AG-dinucleotide preceded by a polypyrimidine tract, consisting of a 15–20-nucelotide pyrimidine-rich region. A long, uninterrupted polypyrimidine tract strengthens the 3′-splice site. It is worth noting that the majority of the HPV16 splice sites contain short sequences of pyrimidines interrupted by purines, with the possible exception of the E2 splice site SA2709 and the late splice site SA5639. Thus, the strength of the 3′-splice sites according to homology to consensus does not predict utilization efficiency, strongly supporting the importance of adjacent, splicing-regulatory RNA enhancer and silencer elements in the control of each HPV16 splice site. Even if there are inherent differences in HPV16 splice site strength, none of the splice sites are allowed to be utilized with a 100% efficiency since all HPV splice sites compete with at least one more splice site on the HPV16 genome (Figure 2).

In conclusion, all HPV16 splice sites are suboptimal and are regulated to respond to cellular differentiation during the course of the HPV life cycle. Unscheduled alterations of HPV splice site efficiency will invariably affect the balanced usage of other HPV splice sites, thereby distorting the relative levels of more than one HPV16 mRNA. Thus, it is of paramount importance to strictly control the usage of all HPV splice sites during the entire HPV life cycle.

HPV16 splice sites SD226, SA409, SA742, SD880, SD1302, SA2582, SA2709, and SA3358 are all used to generate alternatively spliced HPV16 early mRNAs, while SD880, SD1302, SA2582, SA2709, SA3358, SD3632, and SA5639 generate HPV16 late mRNAs (Figure 2). A number of cis-acting regulatory RNA elements and their cognate trans-acting factors that control the HPV16 splice sites or polyadenylation signals have been identified [11,13,16] (Figure 4, Figure 5, Figure 6 and Figure 7). These regulatory RNA elements include splicing activator and silencer elements at the various splice sites, in the early and late 3′-UTR sequences and at early and late polyA signals. Furthermore, there are hot spots for cellular RBPs at HPV16 splice sites SA3358, SD3632, and SA5639 [48] (Figure 5), strongly suggesting that multiple cellular factors control each HPV16 splice site.

#### 4.1.2. HPV16 Splice Sites SD226, SA409, SA526, and SA742

HPV16 splice sites SD226, SA409, SA526, and SA742 are utilized for alternative splicing of mRNAs spanning the E6 and E7 coding region, with the possible exception of SD226 that may be alternatively spliced to downstream 3′-splice sites SA2709 and SA3358 in addition to SA409, SA526, and SA742 (Figure 2). mRNAs that are unspliced in this region encode full-length E6 and E7, whereas mRNAs that are spliced between 226^409 encode E6*I and E7, while mRNAs spliced between 226^526 encode E6*II and E7 (Figure 2). HPV16 mRNAs spliced between 226^742 encode E6^E7 (Figure 2). That the E6 protein is produced from the unspliced mRNA is uncontroversial, since it is the only mRNA that encodes full-length E6. In contrast, the ORF of the major mitotic driver protein E7 is preceded by upstream ORFs of varying length (unspliced, E6*I or E6*II) on all known E7-encoding HPV16 mRNAs. The identity of the “E7 mRNA” is therefore more elusive. One of the splice variants, 226^409 (E6*I) (mRNA B in Figure 2), has been shown to serve as the major E7-producing mRNA [49]. The E6*I protein may also be produced in its own right, since it has been detected in the HPV16-infected CaSki cell line [50] and was shown to function as an antagonist of the full-length E6 protein [51,52,53], thereby contributing an antitumorigenic function [54]. Irrespective of the function of the E6*I protein, it is apparent that 226^409 splicing is an important splicing event since it appears to be required for production of the E7-producing mRNAs. Therefore, the efficiency by which SA409 is used will determine the ratio between the unspliced E6 mRNAs and the 226^409-spliced E7 mRNAs, thereby controlling the relative levels of E6 and E7, two essential proteins in the HPV16 life cycle as well as in HPV16 carcinogenesis. Splicing between SD226 and SA409 is one of the most efficient splicing events of the HPV16 mRNAs, and the E6*I/E7 mRNA is one of the most abundant mRNAs in HPV16-infected cells, as well as in HPV16-driven cancer cells [53]. Lastly, splice sites in the E6- and E7-coding regions such as splice sites SD226, SA409, SA526, and SA742 in HPV16 are unique to high-risk HPVs [55].

The control of these splice sites by cis-acting sequences and trans-acting factors is currently under investigation, and hitherto it has been reported that 226^409 splicing in HPV16 and the corresponding splicing event in HPV 18 (233^416) are negatively regulated by hnRNP A1 that binds to exonic splicing silencers located in the E7-coding region [56,57] (Figure 5). This suppression promotes production of unspliced E6 mRNAs at the expense of the spliced E7 mRNAs and highlights the importance of hnRNP A1 in the control of expression of the two oncogenes E6 and E7. In HPV16, hnRNP A2 interacts with the same splicing silencer to inhibit SA409 but is different from hnRNP A1 in that it redirects splicing to the downstream 3′-splice site SA742 [57] (Figure 5). Thus, both hnRNP A1 and A2 inhibit splicing to SA409, but inhibition by hnRNP A1 promotes production of E6 mRNAs whereas hnRNP A2 does not. The cellular factors that antagonize hnRNP A1 and A2 to enhance splicing to SA409 in HPV16 or SA416 in HPV18, and thereby stimulate E7 mRNA production, are currently unknown.

#### 4.1.3. HPV16 Splice Site SA3358

HPV16 splice acceptor SA3358 is the most commonly used 3′-splice site on the HPV16 genome and is efficiently used during both early and late stages of the HPV16 life cycle. Despite these properties, it conforms poorly to the consensus 3′-splice site (Table 1). The splicing enhancer immediately downstream of SA3358 is present in both HPV16 and HPV18 and interacts with SR proteins (SRSF1, SRSF3, and SRSF9) (Figure 4) [56,58,59,60,61,62,63]. The sequences adjacent to SA3358 also constitute hot spots for multiple RNA-binding proteins, underscoring the complexity of this region and the control of SA3358 (Figure 3 and Figure 5) [48]. SA3358 is used to generate HPV16 early mRNAs that encode E6 and E6 variants E6*I, E6*II, E6^E7, E7, and E1^E4 and are polyadenylated at pAE, while at the late stage it appears to generate one of the major L1 mRNAs that is spliced from SD880 to SA3358, SD3632, and finally to SA5639, and the only known L2 mRNA simply spliced between SD880 and SA3358 (Figure 2). In addition, HPV16 mRNAs initiated at the late promoter and spliced between SD880 and SA3358 are predicted to produce high levels of the E1^E4 protein, a protein that is produced in high quantities prior to induction of L1 and L2 production (Figure 2). The only difference between E1^E4 mRNAs and L1 and L2 mRNAs is that L1 and L2 mRNAs have to bypass pAE, and reach pAL, whereas E1^E4 mRNAs could be polyadenylated at either pAE or pAL (Figure 2).

#### 4.1.4. HPV16 Splice Sites SD3632 and SA5639

Since HPV16 splice sites SD3632 and SA5639 are exclusively involved in HPV16 late L1 mRNA splicing, they are conditionally inactive during the early stage of the HPV16 life cycle and active during the late stage. Even then, only partial activation of L1 mRNA splicing is allowed as complete activation would inhibit L2 mRNA production. Inhibitory RNA elements that suppress HPV16 late splice sites SD3632 and SA5639 have been identified [48,63,64]. Since HPV16 SD3632 is used in a mutually exclusive manner to HPV16 pAE, SD3632 is strongly suppressed by splicing silencer elements to allow for polyadenylation at pAE. This splicing silencer coincides with an RBP-binding “hot spot” [48,63,64] (Figure 6) which includes the hnRNP D family and hnRNP A2/B1-binding motif (two AUAGUA motifs). In addition to the hnRNP D proteins that suppress SD3632 in cancer cell lines, hnRNP L has been shown to bind the “hot spot” at SD3632 and suppress SD3632 [48] (Figure 6). This is of particular interest since interactions of hnRNP L with HPV16 mRNAs was regulated through hnRNP L phosphorylation by the cellular Akt signaling pathway [48]. Since it was shown that inhibition of Akt in cervical cancer cells as well as in keratinocytes activated HPV16 splice site SD3632, one may speculate that Akt plays a significant role in the control of HPV16 splicing regulation (Figure 7).

Splicing silencers at HPV16 3′-splice site SA5639 were mapped to a region downstream of SA5639 in the L1-coding region and were shown to consist of AU-rich sequences that interact with hnRNP A1 [65,66] (Figure 6 and Figure 7). The region downstream of SA5639 is believed to encode multiple RNA elements, including enhancer and silencer elements that regulate SA5639 [65]. Interestingly, this splicing silencer also coincides with a hot spot for RBPs that include hnRNP L [48] (Figure 6 and Figure 7). Similarly to the binding of hnRNP L at SD3632, the binding of hnRNP L at SA5639 was regulated by Akt-dependent phosphorylation [48] (Figure 7). Thus, both HPV16 exclusively late splice sites SD3632 and SA5639 are controlled by the Akt kinase and by multiple cellular RNA-binding proteins such as hnRNP A1, D, and L, of which hnRNP L plays a special role since it is regulated by Akt [48]. The regulation of SD3632 and SA5639 is complex since it contributes to the shift from early to late HPV16 mRNA splicing and therefore has evolved to respond to signals mediated by extracellular stimuli and/or intracellular environment change under the epithelial differentiation. Indeed, “hot spots” located in narrow regions at both SD3632 and SA5639 interacted with multiple RBPs (Figure 6), suggesting that they contribute to the control of SD3632 and SA5639. To elucidate how the activity of these HPV16 RNA-binding proteins changes during mitosis or in differentiated cells should be rewarding.

#### 4.1.5. HPV16 Early and Late 3′-UTR Sequences and Polyadenylation Signals

In addition to the HPV16 splice sites discussed previously, HPV16 polyadenylation signals are also controlled by regulatory RNA elements [11,16]. The HPV16 early polyadenylation signal (pAE) is located downstream of the E5 gene, and all early mRNAs are polyadenylated at pAE (Figure 2). The pAE is exceptional in that it is subject to regulation during the entire HPV16 life cycle. The pAE is efficiently used at the early stage of the HPV life cycle to secure high expression of the HPV16 early mRNAs, but equally important is its role in preventing read-through of transcription into the HPV16 late gene region. Thus, it stands as a guarantor for suppressor of HPV16 late gene expression at the early stage in the HPV16 life cycle. HPV16 and HPV31 pAE is controlled by upstream and downstream sequences [38,39,67,68,69]. Sequences located in the HPV16 L2 ORF downstream of HPV16 pAE positively control the activity of pAE by binding to hnRNP H and CstF-64 via multiple triple-G motifs [39,67] (Figure 4). hnRNP H controls RNA splicing and polyadenylation [70], whereas CstF-64 belongs to the core polyadenylation complex [47].

Upstream sequences of HPV16 pAE also positively control the activity of the pAE via binding to polyadenylation factor Fip1 at uridine-rich sequences in the HPV16 early UTR [68] as well as additional proteins involved in polyadenylation at pAE including polypyrimidine tract-binding protein (PTB), RALYL, and hnRNP C [68,71] (Figure 4). The UTR also controls nearby splices sites by serving as a landing pad for cellular splicing factors to enhance the polyadenylation at HPV16 pAE. Overexpression of hnRNP C has been shown to alleviate suppression of late splice site SD3632 in an HPV16 early 3′-UTR-dependent manner [72] (Figure 7). This effect of hnRNP C increased after activation of the DNA damage response and resulted in activation of HPV16 late splice site SD3632 and inhibition of polyadenylation at pAE [73] (Figure 4, Figure 7, and Figure 8). HPV is known to activate the DDR and to utilize it for HPV DNA replication [28]. Apparently, the activated DDR also recruits RNA-binding proteins to HPV16 that alter HPV16 RNA splicing and polyadenylation to stimulate late gene expression following HPV16 DNA replication [73].

The HPV16 late polyA signal, pAL, is strongly suppressed by negative RNA elements located in the late UTR and that interact with U1snRNP and CUGBP [65,74,75,76], but other proteins have been reported to bind to the UTR as well (Figure 7) [77]. The negative regulatory effect of the late UTR is a conserved property of many HPVs, although it appears that the HPV16 late UTR is particularly inhibitory [65]. Apparently, the suppression of the HPV16 pAL is an important property of HPV16 that probably contributes to the strong suppression of HPV16 late gene expression at the early stage of the HPV16 life cycle.

#### 4.1.6. The 5′-Untranslated Region of HPV16

Regarding the HPV16 5′-UTR region, cellular RNA-binding proteins interacting with this region have not been identified to the best of our knowledge. However, cis-acting regions in the LCR interact with trans-acting transcription factors or DNA replication factors [78,79] that may affect HPV16 RNA processing. It has also been reported that there is a transcription initiation site upstream of the enhancer region in HPV16 LCR [80], supporting the idea that proteins that bind the enhancer region (auxiliary enhancer: AE and keratinocyte enhancer: KE) in the LCR could potentially affect HPV16 RNA processing (discussed below) or translation. This may be an interesting area for future research.

#### 4.1.7. What Do the Various HPV16 mRNAs Produce?

Although it is clearly important to understand how production of each HPV mRNA is regulated, it is also important to understand what each HPV mRNA produces and how—in particular in the light of the ability of HPVs to produce so many alternatively spliced and polycistronic mRNAs. However, translation of HPV16 mRNAs has been investigated only for E6/E7, E5, and E2 mRNAs [81,82,83,84,85]. The E7 ORF is always preceded by an upstream ORF (E6, E6*I or E6*II) on all known E7-encoding mRNAs (Figure 2). Since the scanning model for mRNA translation predicts that upstream ORFs or ATGs inhibit translation of downstream ORFs [86], it is unclear how E7 is translated. It appears that the best E7-producing HPV16 mRNA is spliced 226^409 and encodes E6*I upstream of E7. The altered or shortened upstream E6 ORF may contribute to the higher ability of this mRNA to produce E7 by translation. It has been shown previously that short upstream ORFs are less of an obstacle for translation of downstream ORFs on mRNAs translated by the scanning mechanism [87]. However, other mechanisms for translation may operate on the HPV16 E6/E7 mRNAs [81]. HPV16 E5 is preceded by upstream ORFs and ATGs on all known alternatively spliced mRNAs encoding E5 (Figure 2). Despite the presence of the E5 ORF on a majority of all alternatively spliced HPV16 mRNAs, it was shown that HPV16 E5 was translated primarily from HPV16 mRNAs spliced from SD226 and SA3358 (226^3358) [83] (Figure 2, mRNA F). A recent study showed that HPV16 E2 was efficiently translated from HPV16 mRNAs initiated at HPV16 nucleotide position 670, that corresponds to the transcription initiation site of the HPV16 late promoter, and spliced between SD880 and SA2709 (880^2709) [84] (Figure 2, mRNA J). Two splice sites are located upstream of the E2 ATG in HPV16: SA2582 and SA2709 (Figure 2). HPV16 splice site SA2582 conforms less well to the consensus 3′-splice site than HPV16 SA2709 (Table 1), that is also located closer to the E2 ATG than SA2582 (Figure 2). The E2 mRNAs spliced to SA2582 produced less E2 protein than E2 mRNAs spliced to SA2709 [84]. The role of the HPV16 SA2582 splice site is unclear. Further investigation is required to delineate the translation potential of each HPV16 mRNA.

### 4.2. Are Splicing and Polyadenylation of HPV mRNAs Cotranscriptionally Controlled?

#### 4.2.1. Cotranscriptional Control of RNA Processing

A number of reports suggest that splicing is dramatically affected by changes in transcription [88]. It has become clear that cotranscriptional splicing is spatially and temporally linked to transcription, and that a key player in coordinating transcription with splicing is the RNA polymerase itself. Changes in the kinetics of the RNA Pol II elongation can affect splice site selection in alternatively spliced genes [89,90,91,92]. The regulation in cotranscriptional control of RNA processing includes the following: (i) Pol II elongation rate can be altered by the epigenetic modifications including chromatin modifications, chromatin structure, and nucleosome occupancy, which can form a natural barrier to the transcribing polymerase [93,94]. As a result of change in Pol II elongation rate, the ability of splicing factors to bind to sequences in the nascent RNA can be influenced, as slow Pol II favors the recruitment of specific RBPs that promote exon inclusion or exclusion, whereas a fast Pol II will hamper that recruitment [95,96]. (ii) Phosphorylation of the Pol II C-terminal domain (CTD) at serine 2 and 5 is known to couple transcription and numerous RNA processing events [97,98]. CTD seine 5 phosphorylation of CTD by the basal transcription factor TFIIH [99] precedes serine 2 phosphorylation by transcriptional elongation factor P-TEFb. These phosphorylation events lead to transcriptional elongation [100,101,102]. It has been suggested the CTD interacts directly with RNA splicing factors to recruit them to the nascent transcript [100,101,103]. For example, the splicing factor U2AF65 interacts directly with the phosphorylated CTD [104], and SR proteins have been shown to interact with RNA Pol II CTD and affect pre-mRNA splicing [105]. (iii) Promoters and transcriptional factors may also influence alternative splicing [106], presumably by crosstalk between transcription factors and splicing factors such as hnRNP A1/A2, hnRNP K, hnRNP L, and SRSF1 and SRSF3, and transcription factors that are associated with core promoters [107,108,109,110].

#### 4.2.2. HPV16 RNA Processing May Be Cotranscriptionally Regulated

Epigenetic changes of the HPV16 chromatin have been described [111]. More than one hundred CpG motifs (that could potentially be methylated) are present in noncoding as well as in protein-coding regions of the HPV16 genome. These CpG sites are unevenly distributed over the HPV16 genome [112]. Methylation of CpG sites in the E2, L1, and L2-coding regions appear to increase as high-grade lesions develop in persistently HPV-infected women [112]. Indeed, it was shown that the different CpG methylation pattern on HPV16 genome influenced E2-dependent transactivation activity [113]. In addition, in vitro experiments demonstrated that modifications of HPV16 DNA-associated histones were differently distributed on the HPV16 genomes [114]. Uneven distribution of histone modifications was observed also on integrated HPV18 DNA in HeLa S3 cells [115]. Furthermore, recruitment of cellular RNA-binding proteins hnRNP C and U2AF65 was affected by the state of the HPV16 chromatin following activation of the DDR [73]. Chromatin-organizing CCCTC-binding factor (CTCF) binds to sequences in the HPV E2 ORF of high-risk HPV types [116] to create CTCF-YY1-mediated HPV18 DNA loop formation [117]. Interactions between CTCF bound to the HPV18 E2 ORF with YY1 bound to the promoter/enhancer region in the HPV18 LCR resulted in epigenetic repression of the HPV18 LCR and affected E6/E7 mRNA splicing [116,117].

The activation of the HPV early promoter which is located in the HPV LCR is regulated by a number of cellular transcriptional factors, such as Sp1, AP1, TEF1, NF1, Oct1, YY1, and CDP [78] and by HPV E2 [29]. The activation of the HPV late promoter, which is located in the E7 ORF, is regulated by cellular transcriptional factors such as Skn-1α, C/EBP-α and -β, c-Myb, and NF-1 in differentiated epithelial cells [78,79]. In addition to the two major promoters, there may be additional minor promoters, for example, P14 in the HPV16 LCR [118,119], a cryptic transcription initiation site upstream of promoter-enhancer region in the HPV16 LCR [80]. Multiple promoters have been found in the HPV-31 LCR [40], as well as in bovine papillomavirus type 1, cotton tail rabbit papillomavirus, HPV1, and HPV8 [120,121,122,123]. It has not been investigated if the HPV promoters, or the binding of transcription factors to their regulatory regions, are connected to the control of HPV mRNA splicing. Taken together, epigenetic changes on HPV genome can affect HPV mRNA splicing and polyadenylation. It remains to be seen if HPV promoters and their cognate transcription factors affect HPV RNA splicing and polyadenylation.

### 4.3. mRNA Export

mRNA export is a critical step of gene expression and includes mRNA transport through the interchromatin space, docking at and export through the nuclear pore complex (NPC), and release of the mRNA into the cytoplasm which are modulated by assembly of pre-mRNP complexes and acquisition of export receptors. There are two major export receptors described, NXF1 and CRM1 [124]. The main mRNA export receptor is NXF1/NXT1 heterodimer and it binds to mRNPs via different adapter proteins. The TREX-1 complex component Aly/REF is one such adapter that is recruited to mRNAs in a splicing-dependent manner [125,126,127]. In addition to Aly/REF, three SR proteins (9G8, SRp20, and ASF/SF2) also serve as NXF1 adaptors in a phosphorylation-dependent manner [107,128]. The function of Nxt1 is not fully understood though Nxt1 has shown to enhance the binding of the NXF1–RNA complex to nucleoporins to promote mRNA export [129]. Another mRNA export receptor named CRM1 is utilized for some mRNAs, including ribosomal RNAs and uridine-rich small nuclear ribonucleotide particles (U snRNAs). CRM1 has been shown to be necessary for export of unspliced or partially spliced HIV1 mRNAs by HIV1 Rev [130,131]. CRM1 is not itself an RNA-binding protein and needs adaptor proteins and interactions with Ran GTPase to function. The three cellular RBPs HuR, LRPPRC, and NFX3 [132,133] have been reported to act as adaptors for CRM1 [134].

There is no direct evidence linking particular RBPs to HPV mRNA export. However, it has been shown that HuR binds specifically to the HPV1 late UTR and to the HPV16 late UTR, suggesting that HuR could potentially enhance HPV late gene expression by promoting nuclear export of late mRNA [135,136,137], possibly via a CRM1-dependent pathway. HPV produces a number of incompletely spliced mRNAs that may be inefficiently exported to the cytoplasm (Figure 2), and it has been shown that the inhibitory effect of the HPV1 and HPV16 late 3′-UTRs on gene expression could be overcome by the HIV1 nuclear export factor Rev and the Rev-responsive element (RRE), as well as by the Simian retrovirus type 1 constitutive transport element (CTE) [138,139,140]. These results suggest that the HPV late UTR sequence may be involved in nuclear export. Many cellular proteins that have been shown to interact with HPV mRNAs, including hnRNP A1, A2, D, E, I, and K (Figure 4 and Figure 7), shuttle continuously between the nucleus and cytoplasm [141,142,143] and are involved in mRNA nuclear export, suggesting that they could potentially contribute to HPV mRNA export. Similarly, many SR proteins shuttle and can serve as NXF1 adaptor proteins, i.e., SRSF1 [128], raising the possibility that interactions of HPV mRNAs with SR proteins could potentially contribute to nuclear export of HPV mRNAs.

### 4.4. RNA Stability

mRNA stability or half-life can be regulated by sequence-specific RBPs that bind to regulatory RNA elements and modulate the interaction of the mRNA with the cellular RNA degradation machinery. mRNA decay rate is transcript-specific [144,145] and varies over a range of up to 100-fold [146]. mRNA in mammalian cells is typically longer-lived, with half-lives ranging from less than 20 min to several days [147,148,149,150]. Eukaryotic mRNAs are protected from exonuclease degradation by the 5′ methylated guanosyl cap and the 3′ poly(A) tail, which is coated with poly(A)-binding protein (PABP). In the most common pathway [145], degradation starts with the removal of one or both of these protective structures, including deadenylation and decapping, followed by exonuclease degradation.

The stability of many labile mRNAs is frequently controlled by cis-acting sequences called AREs (AU-rich elements) that are generally located in the mRNA 3′-UTRs. A number of ARE-binding proteins associate with AREs and recruit components of the core degradation machinery, e.g., hnRNP D, TTP, BRF1, TIA-1, TIAR, and KSRP [151,152,153,154,155,156]. On the other hand, Hu proteins (including HuR) are ARE-binding proteins that are believed to stabilize their targets, but the exact molecular mechanism is unknown [157,158].

mRNA instability element has been identified in the L1 and L2 ORFs of high-risk HPV16, but not in the L1 and L2 ORFs of low-risk HPV1 [135]. These RNA instability sequences may contribute to suppression of HPV16 L1 and L2 mRNA production during the early stage of the HPV16 life cycle. However, it remains to be determined if the stability of these mRNAs is subject to regulation or if it is an inherent property of the AU-rich HPV16 mRNAs. ARE elements have been identified in the HPV16 early 3′-UTR [159] and on the HPV1 late 3′-UTR [139,160]. The HPV1 late UTR element is strongly homologous to the ARE elements in the 3′-UTR of the labile c-fos oncogene mRNAs, and HPV1 and c-fos mRNAs interact with the same set of cellular RNA-binding proteins [160]. RNA elements in the HPV16 early 3′-UTR have shown an inhibitory function for its RNA stability, at least in mitotic cells [159], but also have shown to stimulate HPV16 early polyadenylation [68]. Taken together, these results exemplify how increased knowledge of HPV RNA processing may contribute to our understanding of HPV pathogenesis as well as identification of novel targets for therapy to HPV infections or HPV-induced cancer.

### 4.5. m6A Modification on HPV mRNAs

Recent studies demonstrated that RNA methylation—primarily N6-methyladenosine—has an important biological function in the regulation of different cellular processes such as metabolism, embryonic development, and stem cell self-renewal in an RNA-processing-dependent manner [161]. N6-methyladenosine (m6A) is the most abundant internal modification on mRNAs catalyzed by a methyltransferase complex which consists of “writer” proteins METTL3, METTL14, and WTAP, whereas m6A demethylation is catalyzed by “eraser” proteins FTO and ALKBH5 that are both demethylases. m6A is recognized by “reader” proteins such as YTHDF1, YTHDF2, YTHDF3, and YTHDC1 that affect pre-mRNA splicing, mRNA transport, stability and translation [161]. m6A methylation appears to affect protein–RNA interactions in multiple ways. Methylation can perturb the secondary structure of the mRNA, thereby exposing or masking potential RNA-binding motifs [162]. In case of m6A, association with hydrophobic amino acid side chains or low-complexity regions of proteins may assist in solvation of the modified base. The SRSF1, SRSF3, SRSF9, hnRNP A2, hnRNP C, and hnRNP G proteins have been shown to regulate HPV16 pre-mRNA processing [59,60,72,163], and the binding of these proteins to cellular mRNAs has been shown to be affected by m6A modifications [164,165,166,167,168]. One may speculate that m6A modifications of HPV16 mRNAs may modulate binding of these or other proteins to HPV16 mRNAs, thereby contributing to control of HPV16 gene expression. Indeed, a recent study identified m6A modifications on HPV circular RNAs. CircE7 appears to be generated from HPV16 in HPV16-infected cancer cells, including CaSki. m6A-modified CircE7 RNA encodes full-length E7 ORF and it was suggested that it may contribute to efficient E7 translation [169]. Indeed, a number of putative m6A sites are predicted to be present on HPV16 mRNAs (unpublished), further supporting the idea that HPV pre-mRNA processing could be regulated or fine-tuned by m6A modifications.

## 5. The HPV Life Cycle and Ribonucleoproteins

Expression levels and RNA-binding activity of the RBPs are strictly controlled during tissue development, cell differentiation, and other major cellular events. Many diseases, including cancer, have been linked to defects or dysregulation of RBP expression or function [170,171,172]. Dysregulation of RBPs disrupt the transcriptome balance within tumor cells, thereby driving tumorigenicity. Control of expression levels of RBPs includes changes in transcription, mRNA splicing, mRNA stability, and translation efficiency, whereas control of their function is primarily exerted by post-translational modifications (PTM), as discussed below.

### 5.1. Expression Levels of RNA-Binding Proteins Contribute to the Control of HPV mRNA Processing

Expression levels of RBPs can be modulated during the HPV life cycle in response either to epithelial cell differentiation or to the HPV infection itself. It was originally observed that HPV infection altered the expression levels of a variety of RNA-binding proteins in HPV-infected cells in vivo as well as in HPV-positive precancerous cervical lesions and cervical cancer [173]. Recently, HPV16 or HPV18 infection was found to alter the expression of several RBP genes (e.g., *ELAVL2*, *GRB7*, *KHSRP*, and *PTBP1*) in HPV-infected cervical tissue samples and keratinocytes [174]. Among HPV proteins, HPV16 E2 and E6 may influence the cellular transcriptome via transcriptional regulation, directly or indirectly. HPV16 E2 is a transcription factor that has been shown to affect host cellular gene expression, including splicing factor SRSF1 [175] as well as innate immune genes [176,177], cancer-related genes [178], telomerase [179], cell cycle regulators [180] and to target genes of global transcription factors C/EBP-α and -β [181]. HPV16 E6 may indirectly influence transcription of a number of cellular genes via mediating global transcription factors including p53 [182,183], CBP/p300 [184,185], IRF-3 [186], c-Jun [187], ATF-2 [187], or c-Myc [188,189] or via an unknown mechanism [190,191]. In addition to the changes in the cellular transcriptome caused by HPV early proteins, a number of hnRNP proteins [173] and SR proteins [173,175,192] show dysregulated expression in cervical cancer lesions. It appears that the HPV infection itself alters the RNA-processing potential of the HPV-infected cells by modulating expression levels of major splicing factors such as hnRNPs and SR proteins, thereby affecting alternative splicing of HPV mRNAs and cellular mRNAs. Thus, a full understanding of the HPV life cycle and gene expression program requires a detailed understanding of the interactions between HPV and the cellular RNA-processing machinery.

### 5.2. Posttranslational Modifications of RNA-Binding Proteins Contribute to the Control of HPV mRNA Processing

Functional dysregulation of RBPs is caused by abnormal post-translational modifications (PTM) in many tumors. Abnormal PTM of RNPs originates from defects in one or multiple signaling pathways that cause changes in phosphorylation, acetylation, methylation, and/or ubiquitination of proteins [193]. PTM can change the intra- or inter-binding ability of the modified protein and often serves to regulate protein activity, stability, subcellular localization, folding, and/or interactions with itself or other molecules [194,195]. Although it is not yet known if HPV infections affect PTM of RBPs, HPV16 E1^E4 has been shown to bind and inhibit SRPK1, an enzyme that is known to phosphorylate and activate SR proteins [196,197], but direct effects of HPV16 E1^E4 on HPV mRNA processing have not been reported.

In recent years, it has become increasingly clear that the cellular DNA damage response machinery plays a major role in the HPV life cycle. HPV infections activate the cellular DNA damage response (DDR) [28], and HPV utilizes the DDR for replication of the HPV DNA genome [28] (Figure 8). The DDR itself activates a number of protein kinases in signal cascades. It has been shown that DDR factors activated by DNA damage interact with the HPV16 genome and attract hnRNP C and other RBPs that influence HPV16 mRNA splicing and polyadenylation to the HPV16 DNA [73]. One major consequence of the recruitment of RBPs to the HPV16 genome by DDR was an early-to-late switch in the HPV16 gene expression program [73]. This switch was mediated by interactions between hnRNP C and possibly other cellular factors with HPV16 DNA and pre-mRNAs, which resulted in altered HPV16 splicing and polyadenylation in a cotranscriptional manner and induction of HPV16 late genes expression [73,198] (Figure 8). These results also suggest that PTM of hnRNP C or other RBPs might be affected by the DDR [73].

It has previously been shown that changes in PTM of cellular RNA-binding proteins affect HPV16 RNA processing [48] (Figure 8). More specifically, the phosphorylation status of hnRNP L was shown to be affected by Akt (Figure 8). When Akt was specifically inhibited, HPV16 RNA processing was altered in an hnRNP L-dependent manner [48] (Figure 8). The results revealed that inhibition of Akt altered HPV16 mRNA processing and induced HPV16 late gene expression. Interestingly, Akt signaling is inactivated during the stratified epithelial terminal differentiation [199] that occurs concomitantly with changes in HPV16 late mRNA splicing [48]. Furthermore, the Akt kinase is often overly active in cervical cancer cells [200,201], thereby ensuring low or no expression of the highly immunogenic HPV late L1 and L2 proteins, which may contribute to the immune escape of HPV during carcinogenesis. These results connect Akt signaling with control of HPV16 mRNA splicing and the induction of HPV16 late gene expression (Figure 8).

**Figure 8 viruses-12-01110-f008:**
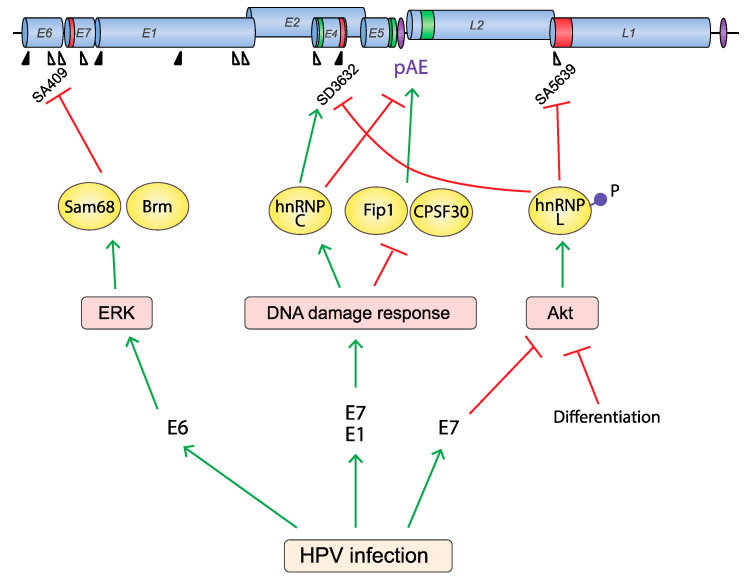
Proteins that regulate HPV RNA processing are controlled by extracellular stimuli and cellular responses. HPV activates the DNA damage response (DDR), the Akt signaling pathway and the ERK signaling pathways upon infection. These activations may affect HPV RNA processing in different ways. The DDR is activated by HPV E1 and E7 by an unknown mechanism to replicate the HPV DNA in differentiating cells [28]. DDR activation also affects HPV mRNA processing by (1) dissociating Fip1 and CPSF30 from HPV16 early UTR, (2) inducing hnRNP C binding to the HPV16 early UTR, thereby alleviating suppression of HPV16 late splice site SD3632 and reducing the activity of HPV16 pAE. As a consequence, the activation of the DDR induces HPV16 late gene expression [73]. The activity of Akt is restricted to epithelial basal cells and Akt is inactivated during epithelial cell differentiation [199] or HPV E7 expression [202], but is often overexpressed in cervical cancer [200]. Inactivation of Akt reduces hnRNP L phosphorylation, which alleviates the silencing of HPV16 late splice sites SD3632 and SA5639. Thus, inhibition of the Akt kinase activates HPV late gene expression at the level of RNA processing [48]. The ERK signaling pathway is activated by HPV E6 [203], and ERK activity affects HPV RNA splicing by controlling E6 exon inclusion via ERK-responding splicing factors Sam68, Brm, and hnRNP A1 [204].

In addition to the Akt signaling pathway, it has been demonstrated that the ERK signaling pathway affects HPV16 alternative splicing of HPV16 E6 mRNAs via ERK-responding RBPs, Brm and Sam68 [204] (Figure 8). Since ERK-signaling is often active in cancer cells, one may speculate that a threshold level of ERK activity is required for production of sufficient quantities of both E6 and E7 in cancer cells. It would be of interest to investigate if pharmacological inhibition of ERK kinases affects HPV16 E6/E7 mRNAs splicing. Taken together, accumulating data indicate that PTM of the RBPs may affect RBP function during the HPV infection, thereby affecting HPV mRNA splicing and contributing to the pathophysiological consequences of the HPV infection. Enzymes that post-translationally modify RBPs to regulate their function may be exploited as targets for therapy to HPV infections or HPV-driven cancers.

### 5.3. Differentiation and HPV RNA Processing

It has been demonstrated that the expression of HPV late transcripts occurred in suprabasal cells of differentiated epithelium in organotypic raft cultures [205,206,207,208] and lesions from patients [209] as well as in keratinocytes induced to differentiate by methyl cellulose [210] or calcium [21]. Furthermore, Mole et al. [211] have shown that splicing factor SRSF1 and its functional regulator SRPK1 were upregulated in HPV16-infected keratinocyte in a calcium-induced differentiation-specific manner. It has been shown that HPV16 late mRNA splicing is modulated by calcium-induced keratinocyte differentiation in combination with Akt kinase inhibition [48]. These studies support the idea that the cell differentiation and signaling pathways influence HPV gene expression and pre-mRNA processing.

### 5.4. Do HPV Proteins Affect HPV RNA Processing?

It has been shown that E2 inhibits HPV early polyadenylation [37]. This allows read-through of the transcription at pAE into the late region of the HPV16 genome and activation of expression of HPV16 late mRNAs that are polyadenylated at the HPV16 late polyA signal. E2 affected the conformation of the polyadenylation complex in vitro, suggesting that E2 interferes with recruitment of the cellular polyadenylation machinery to the HPV16 early polyA signal, pAE [37]. The expression levels of E2 may therefore control the switch from HPV16 early to late gene expression at the level of RNA processing in addition to the inhibitory effects of E2 on the HPV16 early promoter. It has also been shown that overexpression of E2 alters alternative splicing of cellular mRNAs [178], but the mechanism by which this occurs remains to be determined. Furthermore, purified recombinant forms of E2 and E6 showed RNA-binding ability in vitro [212], suggesting that they could potentially affect RNA processing. Recombinant HPV16 E2 and E6 proteins inhibit splicing of HPV RNA templates in vitro [212]. It would be of interest to determine if E2 binds to RNA in living cells as well. E2 is a multifunctional protein that interacts with DNA, proteins, and apparently with RNA. It has also previously been shown that HPV5 E2 interacts with splicing factors of serine-arginine-rich (SR) protein family [213], suggesting that HPV E2 may affect cellular and viral alternative splicing indirectly, by protein–protein interactions. Expression levels of HPV proteins E2, E4, and E6 in HPV-infected cervical lesions depend on CIN grade [209,214,215,216], suggesting that these HPV proteins could potentially affect HPV mRNA processing in a spatiotemporal manner in vivo. Presently, only HPV E2 has been shown to affect RNA processing in living cells. Further research on the role of HPV proteins in the control of HPV mRNA processing is of major interest.

## 6. Conclusions and Perspectives

It is apparent that regulation of HPV RNA processing by HPV–ribonucleoprotein interactions is critical for the control of HPV gene expression, in particular for the temporal production of all alternatively spliced and polyadenylated HPV mRNAs during the entire life cycle, including the switch from the early to the late HPV gene expression program. A number of cis- and trans-acting factors that regulate HPV16 RNA processing have been identified, but additional cis- and trans-acting factors are clearly required for the control of the entire HPV16 gene expression program. To understand the full network of cis- and trans-acting factors that give rise to the HPV transcriptome, a complete identification of these regulatory circuits is required. In addition, many questions remain to be addressed: is HPV pre-mRNA splicing affected by RNA polymerase II kinetics including epigenetic changes on the HPV genome, are RNA modifications contributing to HPV RNA processing, is RNA nuclear export or RNA stability contributing to regulation of HPV gene expression, which PTMs on RBPs affect HPV RNA processing and how are they controlled by intracellular response and/or extracellular stimuli. It is also of importance to perform comparative studies on different HPV types, including cutaneous HPVs, low-risk HPVs as well as all high-risk types, since it could potentially lead to a better understanding of the contribution of regulation of HPV RNA processing to pathogenesis and cancer development. Recent global transcriptomic studies and techniques such as CLIP-seq will help to reveal how RNA processing regulates HPV gene expression. This knowledge may contribute to future development of diagnostic tools for detection of HPV infection as well as for the development of novel drugs to treat HPV infections.

## Figures and Tables

**Figure 1 viruses-12-01110-f001:**
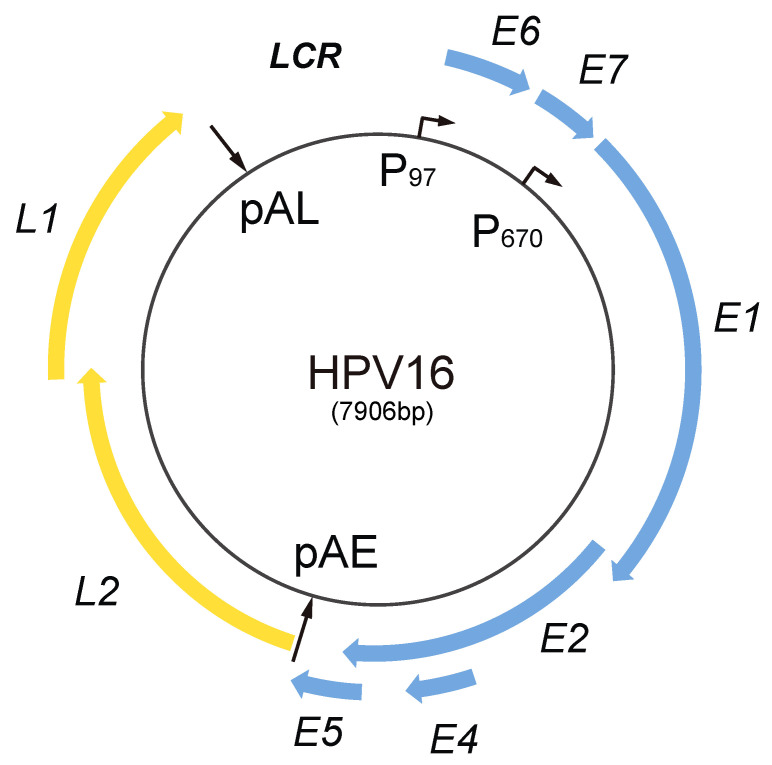
Schematic representation of human papillomavirus type 16 (HPV16) genome. The HPV16 genome is a circular double-stranded DNA molecule of eight kilobases. Blue cylinders represent HPV16 early genes (E1, E2, E4, E5, E6, and E7), and yellow cylinders represent HPV16 late genes (L1 and L2). HPV16 early and late promoters P97 and P670, respectively, and HPV16 early and late polyadenylation signals pAE and pAL, respectively, are indicated. The long control region (LCR) situated between the L1 stop codon and the E6 start codon contains the origin of DNA replication and the HPV16 early promoter/enhancer p97.

**Figure 2 viruses-12-01110-f002:**
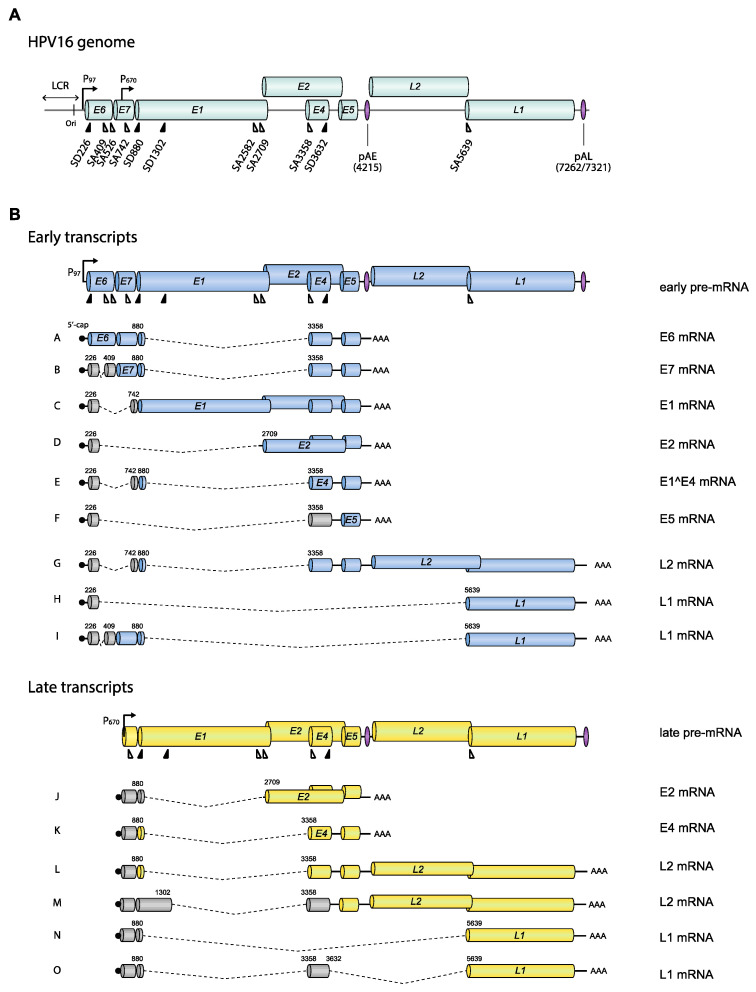
HPV16 genome and transcript map. (**A**) Schematic representation of the HPV16 genome, here depicted as a linearized molecule. Cylinders represent HPV16 early (E1, E2, E4, E5, E6, and E7) and late genes (L1 and L2). HPV16 early and late promoters P97 and P670, respectively, and HPV16 early and late polyadenylation signals pAE and pAL, respectively, are indicated. HPV16 5′-splice sites/splice donors (SD) and 3′-splice sites/splice acceptors (SA) are shown. The long control region situated between the L1 stop codon and the E6 start codon contains the origin of DNA replication and the HPV16 early promoter/enhancer P97. (**B**) A subset of HPV16 mRNAs initiated at the HPV16 early promoter P97 (middle part, blue) or at the HPV16 late promoter P670 (lower part, yellow) are displayed. The transcript map is adapted from the Papillomavirus Episteme website (PaVE) (https://pave.niaid.nih.gov), except the full E1 mRNA (species C), which has not been identified in HPV16-infected cells. Potential coding capacity is indicated to the right of each mRNA.

**Figure 3 viruses-12-01110-f003:**
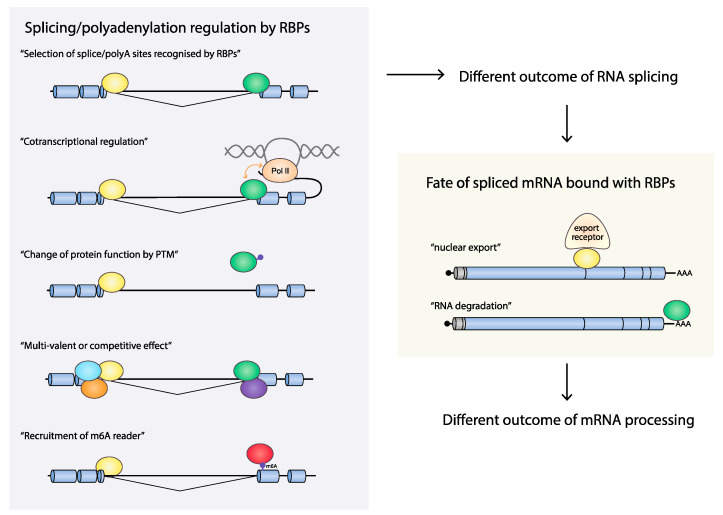
Consequences of RNA-binding proteins on the processing of mRNAs. (Left panel) pre-mRNAs are subject to alternative splicing and/or alternative polyadenylation controlled by interactions of the pre-mRNAs with cellular RBPs. These interactions result in the selection of appropriate splice sites or polyadenylation sites. The interactions between pre-mRNAs and RBPs can be modulated by (1) cotranscriptional regulation by altered by Pol II elongation kinetics or by interactions between Pol II and RBPs, (2) change of protein function by posttranslational modification (PTM), which is regulated by extracellular signaling and/or, for example, cellular responses to HPV infection, (3) multivalent effects and/or competition between RBPs that bind to the same or adjacent cis-acting regulatory RNA elements or hot spots for RNA-binding proteins, (4) m6A modifications on RNAs that affect recruitment of m6A readers that control RNA processing. (Right panel). Some of the RBPs may stick to the mRNAs after RNA splicing and polyadenylation. These RBPs may affect additional RNA processing steps subsequent to splicing, such as nuclear RNA export and RNA degradation.

**Figure 4 viruses-12-01110-f004:**
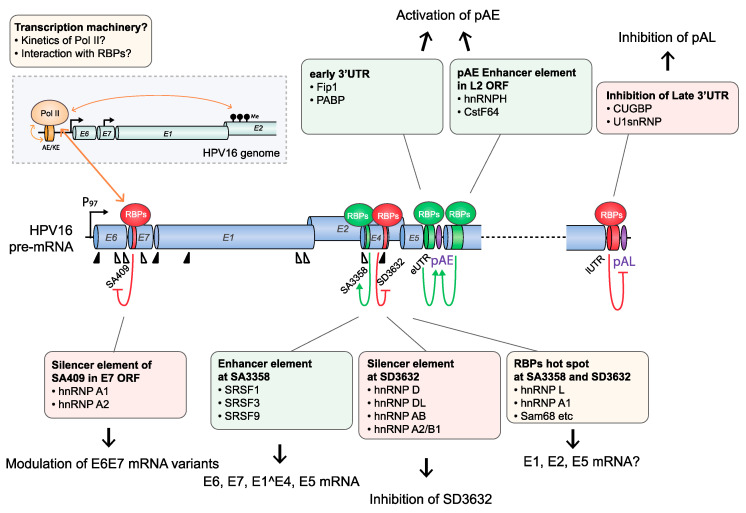
Regulation of HPV16 pre-mRNA splicing and polyadenylation during the HPV16 early life cycle. HPV16 early pre-mRNA splicing and polyadenylation is regulated in a multi-stepwise fashion. Although not yet shown for HPV16, RNA polymerase II (Pol II) kinetics may affect HPV16 splice site selection. Pol II kinetics can be regulated by epigenetic modifications on DNA and/or PTMs of the Pol II C-terminal domain (CTD). In addition, Pol II interacts with various RBPs that may influence RNA splicing. There are multiple cis-acting regulatory RNA elements on HPV16 early mRNAs, indicated here as green cylinders (splicing enhancers) and red cylinders (splicing silencers), that interact with RNA-binding proteins (RBPs) that either activate (green spheres) or suppress (red spheres) splicing or polyadenylation. Proteins that control HPV16 splice sites SA409, SA3358, SD3632 or the HPV16 early and late polyadenylation signals pAE and pAL, respectively, have been identified. Proteins controlling pAE and pAL bind to the early or late untranslated regions, eUTR and lUTR, respectively. Red arrows indicate suppression, and green arrows indicate activation. Proteins that have been identified and shown to control HPV16 splice sites or polyadenylation signals are listed in boxes. See text for more details.

**Figure 5 viruses-12-01110-f005:**
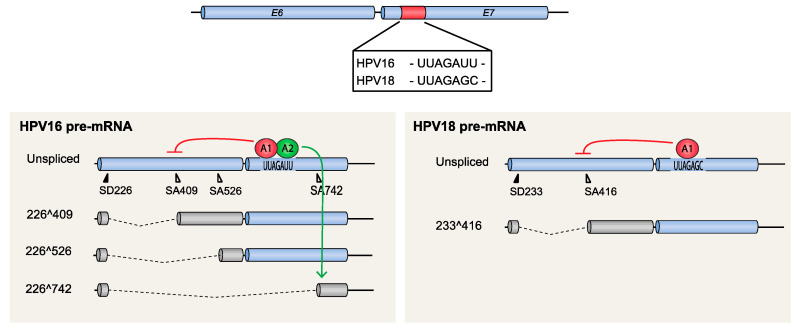
Control of HPV16 and HPV18 E6/E7 mRNA splicing by splicing silencers located in the E7-coding region. Schematic representation of the HPV E6- and E7-coding regions as blue cylinders. The location of a splicing silencer element in the E7-coding region of HPV16 or HPV18 is shown as a red cylinder. The sequence of each splicing silencer element is indicated. The HPV16 splicing silencer interacts with hnRNP A1 and hnRNP A2 to inhibit HPV16 3′-splice site SA409. Inhibition of HPV16 SA409 by hnRNP A1 results in the production of unspliced E6-encoding mRNAs (red arrow), whereas inhibition of HPV16 SA409 by hnRNP A2 results in redirection of splicing to the downstream HPV16 3′-splice site SA742 (green arrow). In HPV18, binding of hnRNP A1 to the splicing silencer results in inhibition of SA416 and production of unspliced E6-encoding mRNAs (red arrow).

**Figure 6 viruses-12-01110-f006:**
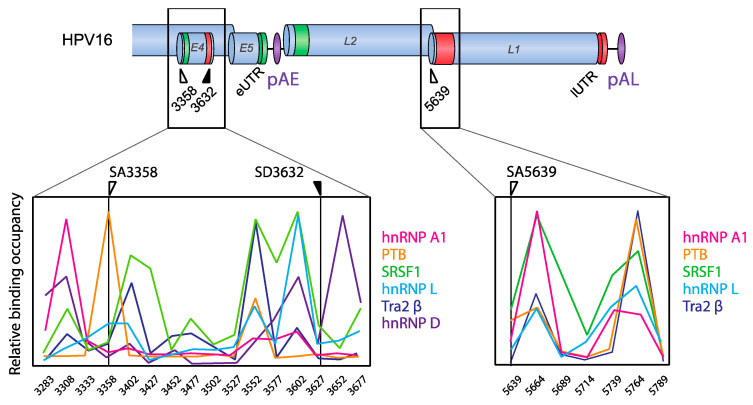
Hot spots for RNA-binding proteins (RBPs) at HPV16 splice sites SA3358, SD3632, and SA5639. Extensive mapping of protein interactions with HPV16 RNAs revealed the existence of hot spots for RNA-binding proteins (RBPs) on HPV16 mRNAs. These “RBP hot spots” were centered around HPV16 splice sites SA3358, SD3632, and SAA5639. Quantitations of the RNA–protein interactions and determination of relative binding occupancy of various RBPs (hnRNP A1, PTB, SRSF1, hnRNP L, Tra2B, and hnRNP D) at these hot spots are shown. The results shown here are adapted from [48].

**Figure 7 viruses-12-01110-f007:**
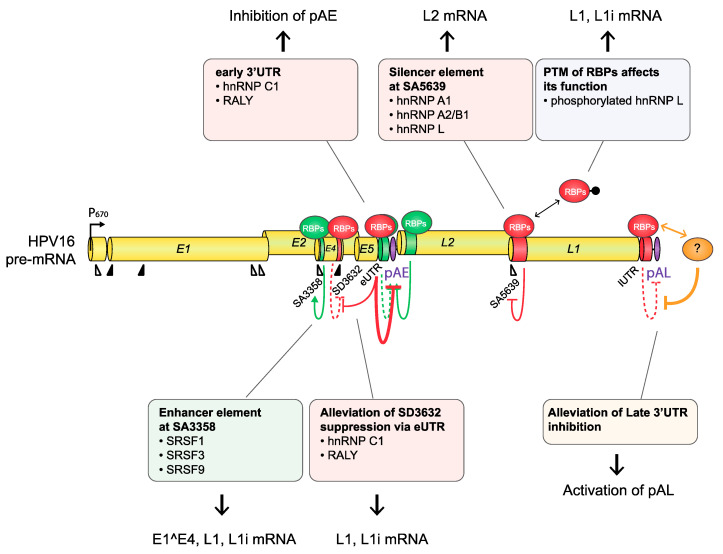
Regulation of HPV16 pre-mRNA splicing and polyadenylation during the HPV16 late life cycle. There are multiple cis-acting regulatory RNA elements on HPV16 late mRNAs, indicated here as green cylinders (splicing enhancers) and red cylinders (splicing silencers), that interact with RNA-binding proteins (RBPs) that either activate (green spheres) or suppress (red spheres) splicing or polyadenylation. Proteins that control HPV16 splice sites SA3358, SD3632, and SA5639 or the HPV16 early and late polyadenylation signals pAE and pAL, respectively, have been identified. Proteins controlling pAE and pAL bind to the early or late untranslated regions, eUTR and lUTR, respectively. Red arrows indicate suppression, and green arrows indicate activation. Dotted lines represent events that must be reversed during switch from early to late gene expression. Proteins that have been identified and shown to control HPV16 splice sites or polyadenylation signals are listed in boxes. See text for more details.

**Table 1 viruses-12-01110-t001:** Sequence of consensus and HPV16 splice donors and acceptors.

**HPV16 Splice Donors**	
**Consensus Sequence**	(C/A)AG_GU(A/G)AGU
SD226	GAG_GUAUAU
SD880	CAG_GUACCA
SD1302	CAG_GUAGAA
SD3632	AAG_GUGAUG
**HPV16 Splice Acceptors**	
**Consensus Sequence**	(C/U)nX(C/U)AG_(A/G)
SA409	GAUUUGUUAAUUAG_G
SA526	AUGUCUUGUUGCAG_A
SA742	CCUUUUGUUGCAAG_U
SA2582	UAAUGCUGGUACAG_A
SA2709	UCCUUUUUCUCAAG_G
SA3358	ACAUCUGUGUUUAG_C
SA5639	AUAUUUUUUUUCAG_A

Horizontal line indicates excision site. Well conserved sequences are indicated in green. The invariable dinucleotides GU and AG of the 5′- and 3′-splice sites, respectively are indicated in red.

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
