# Peer review of "Role of Viral Ribonucleoproteins in Human Papillomavirus Type 16 Gene Expression"

_viruses, 2020, doi:10.3390/v12101110_

Round 1

Reviewer 1 Report

Kajitani and Schwartz present a comprehensive review on the current knowledge of regulation of HPV mRNA transcript processing. The text is generally well written and is accompanied by a rich list of references. The illustrations are well conceived and clearly summarize the detailed information given in the text.

I would recommend an additional very concise section (or paragraph) providing a general overview of the different types/families of RNA binding proteins that interact with HPV mRNA transcripts and their different outcomes in terms of mRNA processing (i.e. the information conveyed by Figure 3). This section should be introduced early in the text to provide a framework to improve readability of the subsequent detailed descriptions of protein-RNA interactions.

Minor comments.

  • Page 6, lines 144-147. The sentence “One apparent consequence…promoter P670” can be delited since it breaks the discussion on transcription factors.
  • Page 7, line 192. Strange sentence, please check.
  • Page 8, line 215. “Consequently “?
  • Page 9, lines 251-253. The sentence “Thus, HPV16 …late splice sites” repeats the same information of the previous sentence.
  • Page 16, lines 462-477. This paragraph includes points (i), (ii) and (iii) that are not really introduced…
  • Page 18, line 562. Full reference missing.

Author Response

Response to Reviewer 1:

1.I would recommend an additional very concise section (or paragraph) providing a general overview of the different types/families of RNA binding proteins that interact with HPV mRNA transcripts and their different outcomes in terms of mRNA processing (i.e. the information conveyed by Figure 3). This section should be introduced early in the text to provide a framework to improve readability of the subsequent detailed descriptions of protein-RNA interactions.

Response 1: Brief introduction of RNA splicing factors was added in the section “3.2 Exons and Introns on HPV16 mRNAs” (Page 6, line 163-167).

  1. Page 6, lines 144-147. The sentence “One apparent consequence…promoter P670” can be delited since it breaks the discussion on transcription factors.

Response 2: The sentence was deleted.

  1. Page 7, line 192. Strange sentence, please check.

Response 3: The sentence was corrected (Page 8, line 215-216)

  1. Page 8, line 215. “Consequently “?

Response 4: The sentence was corrected. Now reads: “Consequently, there are no constitutively active splice sites on HPV16 mRNAs.” (Page 8, line 215-216)

  1. Page 9, lines 251-253. The sentence “Thus, HPV16 …late splice sites” repeats the same information of the previous sentence.

Response 5: The sentence was deleted.

  1. Page 16, lines 462-477. This paragraph includes points (i), (ii) and (iii) that are not really introduced…

Response 6: A sentence to introduce three points were inserted in Page 16, line 461-462.

  1. Page 18, line 562. Full reference missing.

Response 7: The reference was fixed (Page 18, line 562).

Reviewer 2 Report

This manuscript by Kajitani and Schwartz is a comprehensive review about the role of viral ribonucleoproteins that regulate HPV16 gene expression. This review is very well written with over two hundred references to support the literature, and I commend the authors on undertaking this herculean effort. I have one comment:

The authors have clearly stated the differences in early and late gene expression, where late gene expression results in transcripts coding for the capsid proteins needed for virion morphogenesis. Late gene expression and virion synthesis is regulated by signals for differentiation. In the field, HPV synthesis is well studied in organotypic cultures, and in some instances, in context of methylcellulose and calcium regulated cell differentiation that express transcripts for capid proteins. In this review, I was unable to find any strong mention of the role of differentiation on late gene expression and potential role of viral ribonucleoproteins in mediating this process. The authors should add a section on the role of differentiation (raft cultures, methylcellulose, calcium) and how the signaling involved may influence function of viral ribonucleoproteins and late gene expression.

Author Response

Response to Reviewer 2:

The authors have clearly stated the differences in early and late gene expression, where late gene expression results in transcripts coding for the capsid proteins needed for virion morphogenesis. Late gene expression and virion synthesis is regulated by signals for differentiation. In the field, HPV synthesis is well studied in organotypic cultures, and in some instances, in context of methylcellulose and calcium regulated cell differentiation that express transcripts for capid proteins. In this review, I was unable to find any strong mention of the role of differentiation on late gene expression and potential role of viral ribonucleoproteins in mediating this process. The authors should add a section on the role of differentiation (raft cultures, methylcellulose, calcium) and how the signaling involved may influence function of viral ribonucleoproteins and late gene expression.

Response: A section “5.3 Differentiation and HPV RNA processing” was added on Page 22, line 681 to 688.